# Therapeutic potential of a choline-zinc-vitamin E nutraceutical complex in ameliorating thioacetamide-induced nonalcoholic fatty liver pathology in zebrafish

Bingbing Cao[1☯], Jiali Zhou[1☯], Bo Xia[1], Xiaoqing Li[2¤a], Rui Wang[2¤a], Yiqiao Xu[1], Chunqi Li[1¤b]*

1 Hunter Biotechnology, Inc., Hangzhou, Zhejiang, The People's Republic of China, 2 Opella, Shanghai, The People's Republic of China

☯ These authors contributed equally to this work.
¤a Current address: Opella, Shanghai, China
¤b Current address: Hunter Biotechnology, Inc., Hangzhou, Zhejiang Province, China
* 819537524@qq.com

## Abstract

Choline has been proven to be effective in maintaining liver function. However, the effect of choline, in combination with other nutrients, on the improvement of non-alcoholic fatty liver disease (NAFLD) remains unclear. This study aimed to investigate the potential effect of the nutraceutical complex containing choline bitartrate, zinc citrate, and dl-α-Tocopheryl acetate on NAFLD in the zebrafish model. The NAFLD model was induced in zebrafish by administering thioacetamide. Experimental groups were established, including a normal control group, the model control group, the positive control group, the nutraceutical complex intervention group, and the choline bitartrate alone intervention group. The intervention was administered to the zebraf-ish in a water-soluble form, while the positive control group received polyene phos-phatidylcholine at a concentration of 50.0 μg/mL. Notably, the protective effect of the nutraceutical complex against NAFLD is more pronounced than that observed with choline bitartrate supplementation alone. The results of transcriptomics and quanti-tative real-time PCR showed that the potential mechanisms underlying the effects of the nutraceutical complex might involve the upregulation of acacia, *acsl1a*, *fbp2* gene expression, and the downregulation of *tbc1d1* gene expression. These results were further validated by western blotting and overexpression experiments. Our findings indicated that choline bitartrate, zinc citrate, and dl-α-Tocopheryl acetate can help improve NAFLD. The results of this study provide evidence for the application of the nutraceutical complex in the improvement of NAFLD.

**Data availability statement:** All relevant data are within the manuscript and its Supporting information files.

**Funding:** Sanofi funded the study.

**Competing interests:** Xiaoqing Li and Rui Wang are currently employees of Opella. Bingbing Cao, Jiali Zhou, Bo Xia, Yiqiao Xu and Chunqi Li are employees of Hunter Biotechnology, Inc. which performed the study by order of Opella. This does not alter our adherence to PLOS ONE policies on sharing data and materials.

## Introduction

Nonalcoholic fatty liver disease (NAFLD) is the most common liver disease worldwide. With the change in lifestyle and diets, hepatobiliary diseases have risen to one of the leading causes of death and disease worldwide [1]. It is estimated that liver-related diseases result in the deaths of over two million individuals annually, constituting approximately 4% of the total global mortality rate [2]. In the United States, NAFLD affects approximately 30% of the population, while its subtype, nonalcoholic steatohepatitis (NASH), impacts around 5% [3]. The prevalence of NAFLD is strongly correlated with obesity, type 2 diabetes (T2DM), dyslipidemia, and hypertension [4,5]. Despite the high prevalence of NAFLD, the lack of approved pharmacological treatments is primarily due to the insufficient understanding of its underlying mechanisms. In addition, although medical treatments including T2DM drugs (Glucagon-like peptide-1 receptor agonists and Liraglutide), lipid-lowering agents (Statins and Ezetimibe), and antihypertensive medications are reported to have some effect in reducing NAFLD, the global situation is still not optimistic [6–12]. The prevalence, mortality, and healthcare burden of hepatobiliary diseases are steadily increasing. There is an urgent need for more cost-effective and new strategies to prevent and treat hepatobiliary diseases.

Dietary nutrient supplements and natural herbal ingredients may be critical in improving NAFLD [13,14]. Choline, being an indispensable nutrient, undergoes metabolic processes within the liver. Choline is a constituent of mitochondrial membranes and cell membranes [15]. Food sources rich in choline include egg yolks and animal protein sources [16]. Choline is synthesized in the liver and stored there as well. Multiple mouse models with choline-related gene deletions have been employed to elucidate the underlying mechanisms of NAFLD [17]. These alleles exhibit a strong correlation with the advancement and manifestation of NAFLD. It is reported that the administration of choline supplements has been found to confer benefits to individuals diagnosed with NAFLD [18]. Zinc, akin to choline, potentially contributes to the amelioration of NAFLD through its modulation of peroxisome proliferator-activated receptor alpha (PPAR-α) and lipid metabolism, insulin-like growth factor-1 (IGF-1), and insulin activity within cellular systems [19–21]. Zinc deficiency has been linked to the progression of hepatic fibrosis and the severity of NAFLD [22–24]. Besides, Hepatic ZIP14 was increased because of its implications for ER stress adaptation [25]. Vitamin E, functioning as a lipid-soluble antioxidant, prevents the spread of free radicals [26]. Plants synthesize vitamin E in the form of four tocopherols and four tocotrienols. The impact of vitamin E on NAFLD has been investigated in various experimental models [27]. The administration of α-tocopherol through dietary supplementation demonstrated a mitigating effect on liver injury induced by lipopolysaccharide (LPS), while also inhibiting the occurrence of disease-associated conditions such as oxidative stress and inflammation-related pathologies [28]. Additionally, the expression of TGF-β, a key factor in the progression of liver fibrosis, was inhibited by Vitamin E through the action of reactive oxygen species (ROS) [29]. Moreover, the administration of vitamin E has been shown to increase adiponectin levels in both mice and adipocytes through the activation of peroxisome proliferator-activated

receptor gamma (PPAR-γ) deletion, ultimately leading to enhanced insulin sensitivity [30,31]. Vitamin E has been observed to mitigate hepatic inflammation and fibrosis through T cell and macrophage recruitment, as well as polarization of Macrophages/Kupffer cells towards M211. Given that these three dietary supplements may have positive modulatory regulations on NAFLD, it is important to explore the effects of the combination of choline bitartrate, zinc citrate and dl-α-Tocopheryl acetate on improving NAFLD.

Zebrafish are highly effective models in developmental and pharmaceutical research [32]. Due to their genetic manipulability, zebrafish have been extensively employed in investigating liver initiation and embryonic development [33]. Compared to alternative animal models employed in research, zebrafish offer greater convenience in identifying liver phenotypes and measuring diverse physiological and pathological indicators in vivo [34,35]. Numerous zebrafish models of NAFLD have been established to investigate the underlying mechanisms of this condition. Prior research has indicated promising prospects in lipid metabolism [36–38]. Hence, we intend to investigate the potential ameliorative effects of the nutraceutical complex containing choline bitartrate, zinc citrate, and dl-α-Tocopheryl acetate on NAFLD through the zebrafish model.

## Materials and methods

### Zebrafish husbandry

The study was approved by the Experimental Animal Ethics Committee of Hunter Biotechnology, Inc. (approval number IACUC-2022-5337-01) and it is considered that the experiment meets the requirement of animal ethics. After each experiment, all the zebrafish were anesthetized and euthanized with 0.25 g/L tricaine methanesulfonate, which conforms to the American Veterinary Medical Association (AVMA) requirements for euthanasia by anesthetic. The Zebrafish was obtained from Hunter Biotechnology. Inc. (License number: SYXK (Zhejiang) 2022–0004). The zebrafish facility and the laboratory are accredited by the Association for Assessment and Accreditation of Laboratory Animal Care (AAALAC) International (001458). All zebrafish were reared in fish-holding water at a temperature of 28°C, with the following water quality conditions: 200 mg of instant sea salt was added to every 1 L of reverse osmosis water, conductivity ranged from 450 to 550 μS/cm, pH was maintained between 6.5 and 8.5, and hardness ranged from 50 to 100 mg/L CaCO$_3$.

Albino zebrafish with a mutation in the melanin allele were bred using natural pairwise mating. Zebrafish at 3 days post-fertilization (3 dpf) were used to determine the maximum test concentration (MTC) and assess the efficacy of zebrafish non-alcoholic fatty liver protection.

### Antibodies and reagents

A new nutraceutical complex containing choline bitartrate, zinc citrate, and dl-α-Tocopheryl acetate (Essentiale Revital® purchased from Opella Healthcare Technology Co., Ltd.) is prepared in standard dilution water to 2.00 mg/mL mother liquor and stored at 4°C. Choline Bitartrate is treated in the same way. Polyenyl phosphatidylcholine was used as the positive control group, and it was prepared using Dimethyl sulfoxide (DMSO) to 50.0 mg/mL mother liquor and stored at 4°C. The instruments used for the experiment are mainly as following: Anatomical microscope (SZX7, OLYMPUS, Japan); motorized focus continuous magnification fluorescence microscope (AZ100, Nikon, Japan); high-speed frozen centrifuge (Heraeus Fresco17, ThermoFisher, Germany); multi-functional enzyme labeler (Tecan, Spark, Switzerland); 6-well plate (Zhejiang Bellanber Biotechnology Co. China).

Reagents and antibodies used in the experiment mainly included Dimethyl sulfoxide (DMSO, lot number: 20171016, Sinopharm Chemical Reagent Co., Ltd., China); Methyl cellulose (lot number: C2004046, Shanghai Aladdin Biochemical Technology Co., Ltd., China); Fat-soluble diazol dye (lot number: SHBN4926, Sigma, USA); Thioacetamide (TAA, lot number: BCBV3031, Sigma, Switzerland); GPT Kit (ELISA) (lot number: 20220517, Nanjing Jiancheng Biological Research Institute, China); GOT Kit (ELISA) (lot number: 20220516, Nanjing Jiancheng Biological Research Institute, China); PBS buffer (Cat No. BL601A, Biosharp, China); 4% tissue cell fixative (lot number: 20210828, Beijing Solabao Technology Co.

Ltd., China); FastKing cDNA First Strand Synthesis Kit (Degenome) (lot number: X1213, Tiangen Biochemical Technology (Beijing) Co., Ltd., China); Universal RNA Extraction TL Kit C (lot number: TL2204001643C, Foshan Aowei Biotechnology Co., Ltd., China); SDS-PAGE Gel Preparation Kit (lot number: 7E661H2, Vazyme, China); BCA Protein Concentration Assay Kit (lot number: AR1189, BOSTER, China); Western Quick Transfer Solution (10X) (lot number: P0572-2L, Beyotime, China); ECL Chemiluminescence Kit (lot number: G2014 -50ML, Service, China); QuickBlock™ Western Blocking Solution (lot number: P0252-500ml, Beyotime, China); Anti-β-Actin Antibody (lot number: BM0627, BOSTER, China); Primary Antibody Rabbit FPB2 Polyclonal Antibody (lot number: MBS320877, MyBioSourc, USA); Primary Antibody Acetyl-CoACarboxylas (lot number: CY5575, Shanghai Bowan Biotechnology Co., Ltd., China); Primary Antibody Rabbit ACSL1 Polyclonal Antibody (lot number: MBS320116, MyBioSourc, USA); primary antibody Rabbit TBC1D1 Polyclonal Antibody (lot number: MBS320992, MyBioSourc, USA); secondary antibody Goat Anti-Rabbit IgG (H&L) Biotin (lot number: BA1054, BOSTER, China).

## Non-alcoholic fatty liver model

Thioacetamide was used to induce Melanin allele mutant translucent Albino strain of zebrafish to construct the NAFLD model [39]. Thioacetamide exerts a direct hepatotoxic effect. Upon ingestion, it undergoes metabolic conversion to TAA thioredoxin via the cytochrome P450 mixed-function oxidase system within hepatocytes. This metabolite disrupts intracellular RNA transfer processes, impairing protein synthesis and enzyme activity. Additionally, it enhances DNA synthesis and mitotic activity in hepatocyte nuclei, ultimately facilitating the progression of cirrhosis. It also activates hepatocyte phospholipase A2, destroys hepatocyte membrane, and forms enteric endotoxemia, leading to extensive hepatocyte destruction and a significant increase in ALT and AST. Thioacetamide induced hepatocyte apoptosis in small doses, and large doses led to lipid oxidation and central lobular necrosis. Thioacetamide induced zebrafish, which resulted in hepatocellular swelling, increased intrahepatic lipid droplet deposition, fatty liver formation, and darkened liver staining.

## Maximum tolerated concentration determination

Three days post-fertilization, albino zebrafish were selected with a mutation in the melanin allele and placed in a 6-well plate with 30/well zebrafish. Each well was treated with the respective sample (125μg/mL, 250μg/mL, 500 μg/mL, 1000μg/mL, 2000μg/mL). Normal control and model control groups were also included, with a volume of 3 mL per well. Except for the normal control group, the remaining experimental groups were treated with TAA dissolved in water to establish a zebrafish NAFLD model. The treatment continued at 28°C until 5 days post-fertilization, and the Maximum tolerated concentration (MTC) of the nutraceutical complex containing choline bitartrate, zinc citrate, and dl-α-Tocopheryl acetate for the protective efficacy against non-alcoholic fatty liver in the zebrafish model was determined.

## Determination of hepatic glutathione and glutamic acid aminotransferase activity

Three days post-fertilization albino zebrafish with a mutation in the melanin allele were randomly selected and placed in a 6-well plate, with 30/well zebrafish in the experimental groups. The corresponding samples (125μg/mL, 250μg/mL, 500μg/mL of the nutraceutical complex equivalent to the concentration of 81.0μg/mL, 162μg/mL, 316μg/mL of choline bitartrate) were dissolved in water and administered to each well. The positive control zebrafish were given 50.0 μg/mL of polyenyl phosphatidylcholine. Normal control and model control groups were also included, with a volume of 3 mL per well. Except for the normal control group, the remaining experimental groups were treated with thioacetamide dissolved in water to establish a zebrafish non-alcoholic fatty liver model. Three parallel experiments were conducted. The treatment continued at 28°C until 5 days post-fertilization. Samples were collected, and the activities of aspartate aminotransferase (AST) and alanine aminotransferase (ALT) in zebrafish tissues were measured using enzyme-linked immunosorbent assay (ELISA).

## Hematoxylin and Eosin staining, and Oil Red O staining

Zebrafish from each group were fixed in 4% paraformaldehyde (PFA) at 4°C for 24 hours, embedded in paraffin, and cut into 4 µm sections. The specimens were deparaffinized, rehydrated, stained with Hematoxylin and Eosin (H&E), dehydrated, and then retrieved and sealed for routine histological examination. Finally, H&E-stained sections were captured using a dissecting microscope (SZX7, OLYMPUS, Japan).

Ten zebrafish were randomly selected for Oil Red O staining from each experimental group. The data were acquired using NIS-Elements D 3.20 advanced image processing software to analyze the intensity of zebrafish liver fat staining, and the results of the statistical analysis of this index were used to evaluate the efficacy of the samples' protection against non-alcoholic fatty liver disease.

## Liver fat transfer efficacy evaluation

Zebrafish of 3 dpf melanin allele mutation Albino strain were randomly selected in 6-well plates, and 30 zebrafish were treated in each well (experimental group). The samples were given in water, the positive control polyenphosphatidyl-choline 50.0 µg/mL, and the normal and model control groups were set up simultaneously, and the volume of each well was 3 mL TAA was given in water to each experimental group except the normal control group to establish the zebrafish NAFLD model. 28°C and continued the treatment for 24 h; each group was injected with fluorescent fat in the liver. After the treatment was continued at 28°C for 24 h, 10 zebrafish were randomly selected from each experimental group and photographed under a fluorescence microscope. The data were collected using NIS-Elements D 3.20 advanced image processing software to analyze the fluorescence intensity of zebrafish liver fat and to evaluate the efficacy of the samples' hepatic fat transfer by the statistical analysis results of this index.

## Quantitative real-time PCR

Zebrafish from the Albino strain with the melanin allele mutant at 3 dpf were randomly chosen and placed in a 6-well plate. Each well contained 30 zebrafish, constituting the experimental group. Water-soluble samples were administered, with a volume of 3 mL per well. Additionally, a normal control group and a model control group were established simultaneously, also with a volume of 3 mL per well. The experimental groups, excluding the normal control group, were administered TAA in a water solution to induce the zebrafish NAFLD model. Three parallel experiments were conducted, wherein zebrafish samples were collected from each group and the total RNA was extracted using the RNA rapid extraction kit. The concentration and purity of the total RNA were determined using a UV-visible spectrophotometer. A total of 2.00 µg of total RNA was obtained from the zebrafish samples, and cDNA was synthesized according to the instructions of the cDNA first-strand synthesis kit, resulting in 20.0 µL of cDNA. The expression of *β-actin*, *acaca*, *acsl1a*, *fbp2*, and *tbc1d1* genes was quantified using q-PCR, with *β-actin* serving as the internal reference for gene expression analysis. The RNA expression levels of the genes *acaca*, *acsl1a*, *fbp2*, and *tbc1d1* were also determined.

## Western blot analysis

The zebrafish samples from each experimental group were collected, and the total protein was extracted using a tissue protein extraction reagent. The protein concentration of each experimental group was determined using the BCA protein concentration assay kit. Protein expression of ACACA, ACSL1, TBC1D1, and FBP2 was detected using the western blot technique. The gray value of the target band was analyzed using the Image J software processing system. The relative expression levels of ACACA, ACSL1, TBC1D1, and FBP2 proteins were determined using β-actin as an internal reference for protein expression.

## Transcriptomics

30/well zebrafish of the 3 dpf melanin allele mutant Albino strain was randomly selected and placed in a 6-well plate. They were treated with the nutraceutical complex at a concentration of 500 µg/mL and Choline Bitartrate at 316 µg/mL in

aqueous solution. Normal and model control groups were also set up, each with a volume of 3 mL per well. Except for the normal control group, the other experimental groups were treated with TAA in water solution to establish a zebrafish model of NAFLD. Three biological replicate experiments were set up in parallel. After 48 hours of treatment at 28°C, the fish were washed with ultrapure water to remove the drug, immediately transferred to 1.5 mL centrifuge tubes (30 fish/tube), and the liquid was aspirated. The tubes were placed in liquid nitrogen for 3 minutes and stored at -80°C for subsequent transcriptomic analysis.

The main steps of constructing and sequencing the DNBSEQ eukaryotic strand-specific transcriptome library include: 1) Total RNA extraction and quality detection of the collected samples; 2) DNase I digestion of the qualified Total RNA samples; 3) Enrichment of mRNA using Oligo(dT) magnetic beads for the digested Total RNA samples; 4) mRNA fragmentation; 5) Synthesis of the first strand of cDNA: Fragmented mRNA is used for the synthesis of the first strand of cDNA using random primers; 6) Synthesis of the second strand of cDNA using dUTP instead of dTTP; 7) End repair, A-tailing, and adapter ligation of the amplified cDNA; 8) PCR amplification of the ligated products, followed by digestion of the second strand template marked with U using UDG enzyme; Recovery of PCR products; 9) Library quality detection; 10) PCR product circularization: After denaturing the PCR products into single strands, circularization is performed to obtain a single-stranded circular DNA library. After digestion of the linear DNA molecules that were not circularized, the final library is obtained; 11) On-machine sequencing: Single-stranded circular DNA molecules are replicated by rolling-circle amplification, forming DNA nanoballs (DNBs), and then subjected to on-machine sequencing. Differential genes are based on the criteria of $|log2FC| > 1$ and Q-value $< 0.05$.

### Synthesis of overexpression plasmids

In this experiment, the acsl1a overexpression plasmid was synthesized by using pcDNA3.1(+) as a vector, and the target gene was directly synthesized on it by double digestion with NheI-EcoRI, and the concentration of the synthesized plasmid was 2,283 ng/µL.

### Evaluation of acsl1a-mRNA validity verification

We randomly selected single-cell stage melanin allele mutant Albino strain zebrafish in 6-well plates and treated 30 zebrafish in each well (experimental group). Zebrafish were injected with acsl1a-mRNA 0.571 ng/tail, and a normal control group was set up simultaneously, with a volume of 3 mL in each well. Three experiments were set up in parallel. After 5 days of treatment at 28°C, the total RNA of zebrafish in each group was extracted using Universal RNA Extraction TL Kit C, and the total RNA concentration and purity were determined using a UV-visible spectrophotometer. The concentration and purity of total RNA were determined using a UV-Vis spectrophotometer. The relative RNA expression of acsl1a gene was calculated using β-actin as an internal reference for gene expression.

### Statistical analysis

All statistical analyses were performed using GraphPad Prism 9.0 and SPSS 26.0 software. The data results were expressed as mean ± Standard error (SE), and $p < 0.05$ indicated statistically significant differences.

## Results

### The MTC of the nutraceutical complex containing choline bitartrate, zinc, and vitamin E

As shown in the S1 Table in S1 File, the maximum tolerated concentration (MTC) of the nutraceutical complex containing choline bitartrate, zinc, and vitamin E for the protective effect against non-alcoholic fatty liver is 500 µg/mL. The nutraceutical complex and choline bitartrate both protect against non-alcoholic fatty liver disease. Smaller values of liver fat fluorescence intensity indicate less fat accumulation, more being transported, and better transport efficacy. With increasing

concentrations, the protective effect of the nutraceutical complex was significantly superior to that of the intervention group with choline alone ($p<0.05$). However, the protective effects of the nutraceutical complex at a concentration of 125 µg/mL show no statistically significant difference compared to choline bitartrate alone at a concentration of 81.0 µg/mL ($p > 0.05$) (Table 1, Fig 1).

## The nutraceutical complex containing choline bitartrate, zinc, and vitamin E reduced ALT and AST enzyme activities in zebrafish NAFLD model

The nutraceutical complex and choline bitartrate all exhibit protective effects against non-alcoholic fatty liver disease. The main effect is the reduction in alanine transaminase (ALT) and aspartate aminotransferase (AST) activity when compared with choline bitartrate alone group ($p<0.001$) (S2 and S3 Tables in S1 File).

## Histopathology of liver sections

As shown in Fig 2, the liver cells in the model control group showed loose structure with numerous lipid vacuoles compared to the normal control group. In addition, a significant reduction in lipid vacuoles was observed in the positive control group compared to the model control group. The liver structure in the groups treated with the nutraceutical complex at concentrations of 125, 250, and 500 µg/mL, and choline bitartrate alone at concentrations of 81.0, 162, and 316 µg/mL indicated a reduction in lipid vacuoles when compared to the model control group. These results suggested a potential hepatoprotective effect of the nutraceutical complex and choline bitartrate.

## Hepatic fat transfer efficacy

The nutraceutical complex containing choline bitartrate, zinc, and vitamin E and choline bitartrate alone affected hepatic lipid transport. When the content of choline bitartrate is equivalent, the nutraceutical complex at concentrations of 125, 250, and 500 µg/mL showed better hepatic lipid transport effects compared to choline bitartrate alone at concentrations of 81.0, 162, and 316 µg/mL (Table 2 and Fig 3).

**Table 1.  The intensity of hepatic lipid staining of the protective effects of nutraceutical complex against non-alcoholic fatty liver.**

| Group | concentration (µg/mL)# | Intensity of hepatic lipid staining (pixels)* |
|---|---|---|
| Normal control group (n = 10) | – | 8556 ± 395[a] |
| Model control group (n = 10) | – | 14151 ± 543 |
| Positive control group (n = 10) | 50.0 | 11362 ± 578[a] |
| Nutraceutical complex containing zinc, choline bitartrate, and vitamin E (n = 10) | 125 | 13633 ± 742 |
| | 250 | 10231 ± 411[a] |
| | 500 | 9200 ± 384[a] |
| Choline bitartrate (n = 10) | 81.0 | 13794 ± 537 |
| | 162 | 11532 ± 402[a,b] |
| | 316 | 11155 ± 402[a,c] |

#In the positive control group, the concentration represents the concentration of polyene phosphatidylcholine. In addition, the nutraceutical complex group containing zinc, choline bitartrate, and vitamin E and the choline bitartrate group each had three concentration levels.

*Data are presented as mean ± SE.

[a]Compared with the control group of the model, $p<0.05$.

[b]Compared with nutraceutical complex containing Zinc, Choline, and Vitamin E at 250 µg/mL, $p<0.05$.

[c]Compared with nutraceutical complex containing Zinc, Choline, and Vitamin E at 500 µg/mL, $p<0.05$.

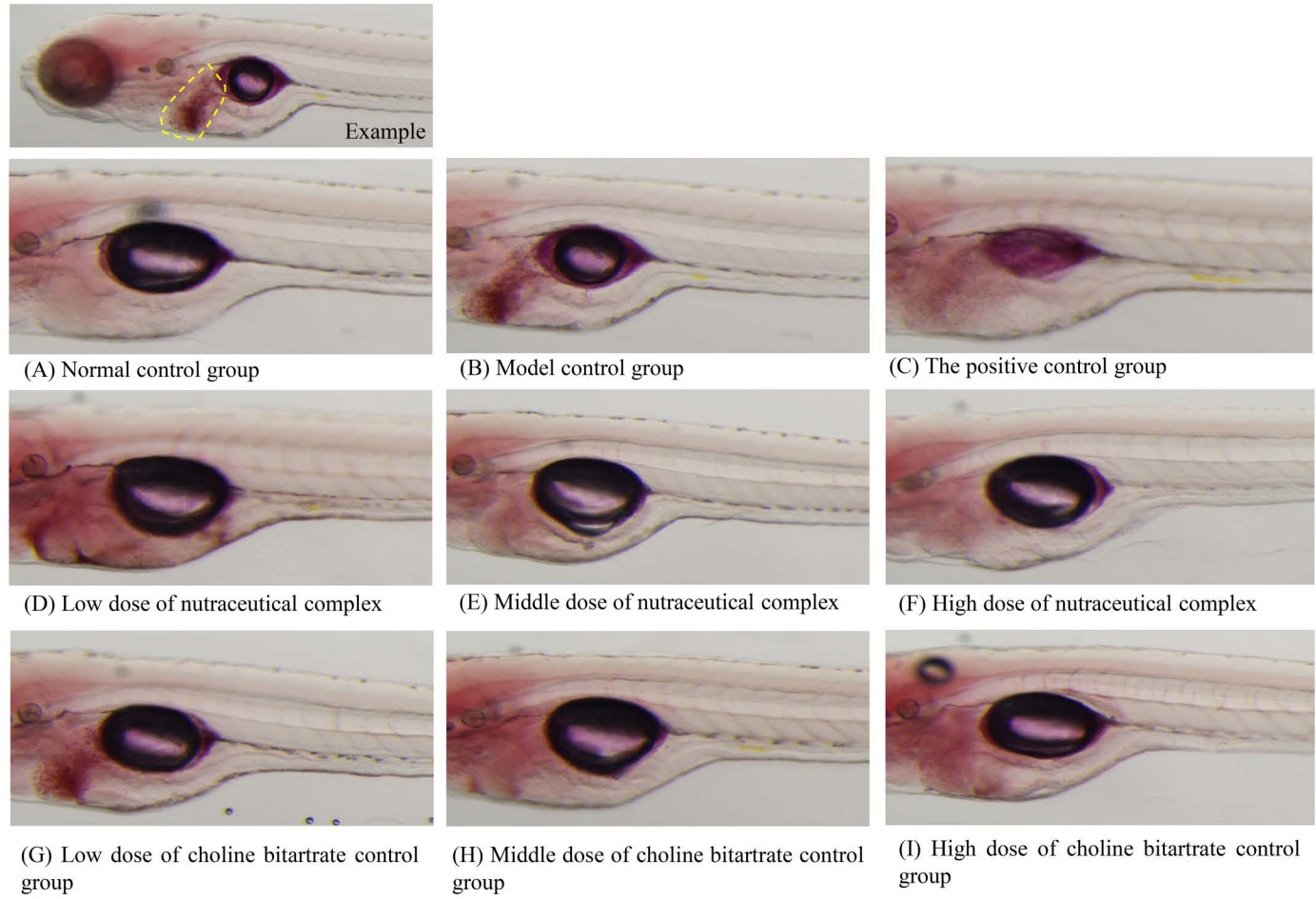

**Fig 1. Intensity of zebrafish liver fat staining after treatment with nutraceutical complex.** (A) Normal control group; (B) Model control group; (C) The positive control group: polyunsaturated phosphatidylcholine 50.0 μg/mL; (D) Low dose of nutraceutical complex: 125 μg/mL; (E) Middle dose of nutraceutical complex: 250 μg/mL; (F) High dose of nutraceutical complex: 500 μg/mL; (G) Low dose of choline bitartrate control group: 81.0 μg/mL; (H) Middle dose of choline bitartrate control group: 162 μg/mL; (I) High dose of choline bitartrate control group: 316 μg/mL. The location of the liver is marked with a yellow dotted line in the example figure.

## Transcriptomics

147 genes were found to be significantly up-regulated in the Model-vs-Control comparison group, while 168 genes were significantly down-regulated. In the Functional formulation-vs-Model comparison group, 170 genes were up-regulated and 131 genes were down-regulated. Additionally, the Choline bitartrate-vs-Model comparison group showed significant up-regulation of 292 genes and significant down-regulation of 185 genes. Moreover, 64 genes exhibited differential expression in both the Model-vs-Control and Functional formulation-vs-Model comparison groups, and there were a total of 71 genes that exhibited overlap between the Model-vs-Control and Choline bitartrate-vs-Model comparisons (S1 Fig in S1 File).

## GO annotation and pathway enrichment analysis of differential genes

Within the realm of biological processes, the three primary genes annotated by Choline to the secondary classification function of GO are cellular process, biological regulation, and regulation of biological process. In terms of molecular

**Fig 2. Liver tissue structure in zebrafish after treatment with nutraceutical complex.** (A) Normal control group; (B) Model control group; (C) The positive control group: polyunsaturated phosphatidylcholine 50.0 μg/mL; (D) Low dose of nutraceutical complex: 125 μg/mL; (E) Middle dose of nutraceutical complex: 250 μg/mL; (F) High dose of nutraceutical complex: 500 μg/mL; (G) Low dose of choline bitartrate control group: 81.0 μg/mL; (H) Middle dose of choline bitartrate control group: 162 μg/mL; (I) High dose of choline bitartrate control group: 316 μg/mL. The black dashed box indicates the observed area in the example picture, and the red arrow points to vacuolar degeneration in the model control group.

functions, binding and catalytic activity exhibit a higher number of gene involvements. The GO annotation analysis of the choline bitartrate differential gene are shown in S2 Fig in S1 File. And GO annotation analysis of functional formulation differential genes are presented in S3 Fig in S1 File.

Specifically, Choline is involved in three cellular processes, two environmental information processing pathways, one genetic information processing pathway, eleven human diseases, seven metabolic pathways, and ten biological systems. The enrichment bubble diagram of the KEGG Pathway is depicted in S4 Fig in S1 File.

Based on conducting KEGG Pathway and GO enrichment analyses, it has been determined that the enrichment of Choline is more pronounced in cholinergic synapse (Chemical carcinogenesis) and choline metabolism (Choline metabolism), as well as in fructose 1,6-bisphosphatase 1 activity (fructose 1,6-bisphosphate 1-phosphate activity), with a higher number of associated genes. The KEGG Pathway and GO enrichment pathways that exhibit pronounced enrichment in the endoplasmic reticulum (ER) and encompass a greater number of genes include Fatty acid biosynthesis, Fatty acid metabolism, AMPK Signaling pathway, and fructose 1,6-bisphosphate 1-phosphate activity. The pathways and differential genes involved in the functional formulation intervention group are summarized in S4 Table in S1 File.

## Effects of functional formulation on genes and proteins associated with NAFLD

The total RNA of zebrafish was extracted after treatment with the nutraceutical complex, and the concentration of RNA and the ratio of A260/A280 were measured with a UV-visible spectrophotometer. The ratio of A260/A280 was between 1.8–2.2, indicating that the extracted the quality of fish total RNA is good and can be used in subsequent q-PCR experiments. Table 3 presented the detailed primer sequences.

Overall, the nutraceutical complex containing choline bitartrate, zinc and vitamin Ehas the protective effect on NAFLD, which is manifested in the up-regulation of *acaca*, *acsl1a*, *fbp2* gene expression and down-regulation of *tbc1d1* gene expression (Table 4). The protein expression of Western blotting showed that the nutraceutical complex has the protective effect of NAFLD, which is specifically manifested as up-regulating the relative expression of TBC1D1 protein and down-regulating the relative expression of ACACA, ACSL1 and FBP2 protein (Fig 4 and S5 Table in S1 File).

## Evaluation of acsl1a-mRNA validity verification

Under the conditions of this experiment, the Maximum detectable dose (MTD) of acsl1a-mRNA to normal zebrafish was 0.571 ng/tail (S6 Table in S1 File). At the end of the experiment, total RNA of zebrafish was extracted, and the

**Table 2. Liver fat transport efficacy in zebrafish after treatment with nutraceutical complex.**

| Group | concentration (µg/mL) | Intensity of hepatic lipid fluorescence (Pixels)[#] |
|---|---|---|
| Normal control group (n = 10) | – | 395471 ± 25071[a] |
| Model control group (n = 10) | – | 935353 ± 38267 |
| Positive control group (n = 10) | 50.0 | 618318 ± 26666[a] |
| nutraceutical complex containing zinc, choline bitartrate, and vitamin E (n = 10) | 125 | 749977 ± 29238[a] |
| | 250 | 535005 ± 27156[a] |
| | 500 | 483228 ± 33055[a] |
| Choline bitartrate (n = 10) | 81.0 | 874488 ± 38250[b] |
| | 162 | 676213 ± 30144[a,c] |
| | 316 | 634717 ± 41524[a,d] |

[#]Data are presented as mean ± SE.

[a]Compared with the control group of the model, $p < 0.05$.

[b]Compared with the 125 µg/mL dosage of nutraceutical complex, $p < 0.05$.

[c]Compared with the 250 µg/mL dosage of nutraceutical complex, $p < 0.01$.

[d]Compared with the 250 µg/mL dosage of nutraceutical complex, $p < 0.05$.

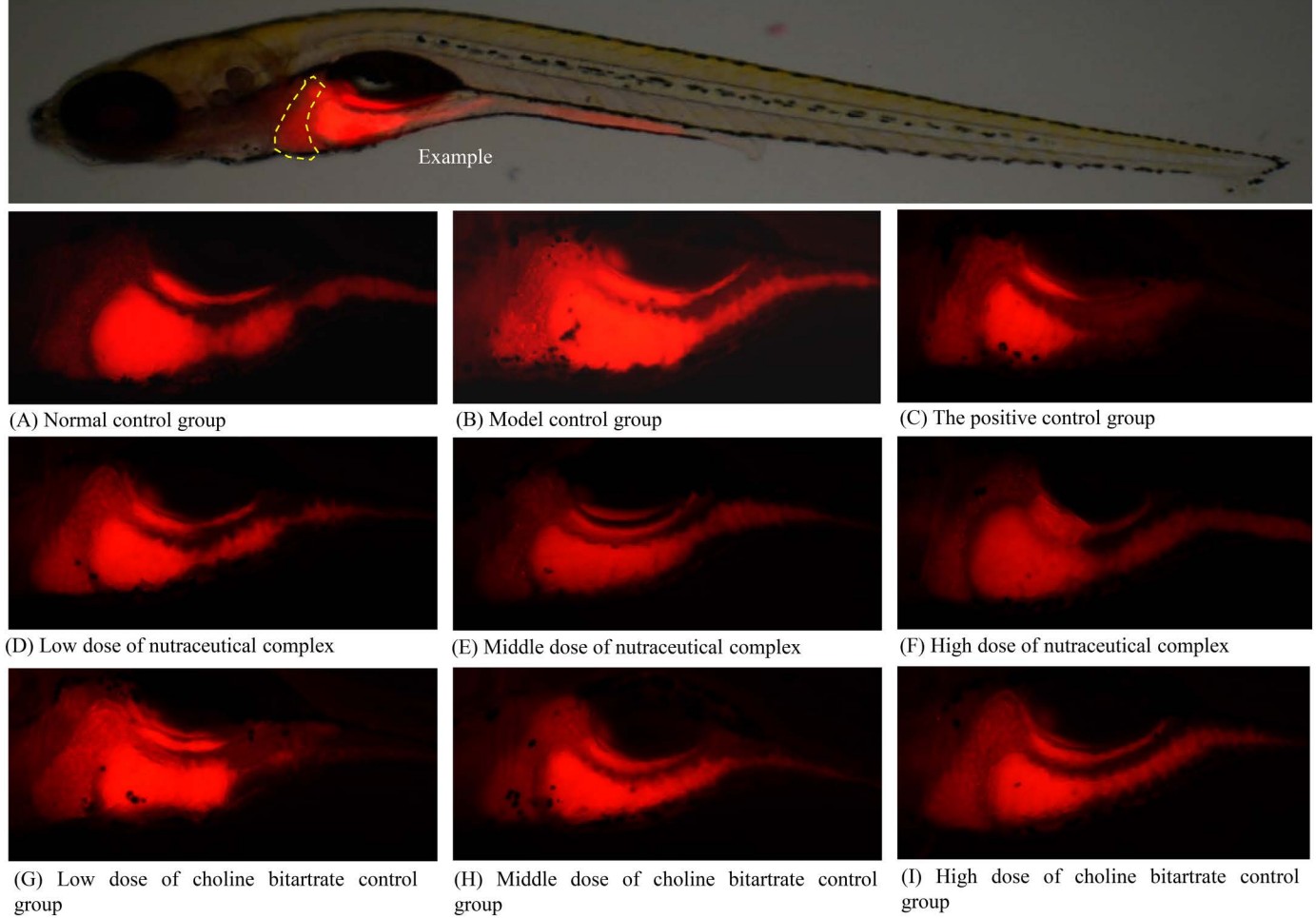

(A) Normal control group
(B) Model control group
(C) The positive control group
(D) Low dose of nutraceutical complex
(E) Middle dose of nutraceutical complex
(F) High dose of nutraceutical complex
(G) Low dose of choline bitartrate control group
(H) Middle dose of choline bitartrate control group
(I) High dose of choline bitartrate control group

**Fig 3. Fluorescence intensity graph of zebrafish liver fat after treatment with nutraceutical complex.** (A) Normal control group; (B) Model control group; (C) The positive control group: polyunsaturated phosphatidylcholine 50.0 µg/mL; (D) Low dose of nutraceutical complex: 125 µg/mL; (E) Middle dose of nutraceutical complex: 250 µg/mL; (F) High dose of nutraceutical complex: 500 µg/mL; (G) Low dose of choline bitartrate control group: 81.0 µg/mL; (H) Middle dose of choline bitartrate control group: 162 µg/mL; (I) High dose of choline bitartrate control group: 316 µg/mL. Yellow dashed line indicates the liver.

concentration of RNA and the A260/A280 ratio were determined by UV-Vis spectrophotometer (S7 Table in S1 File), and the A260/A280 ratios were all between 1.8–2.2, indicating that the total RNA of zebrafish obtained by the extraction was of better quality, and it could be used for the subsequent q-PCR experiments. The primer sequences are shown in S8 Table in S1 File.

In this study, we found that injection of acsl1a-mRNA upregulated acsl1a gene expression (Fig 5). In addition, the nutraceutical complex based on choline has the protective effect on non-alcoholic fatty liver in zebrafish injected with acsl1a gene overexpression plasmid acsl1a-mRNA (Fig 6 and S9 Table in S1 File).

## Discussion

Dietary nutrient supplements and natural herbal ingredients may benefit NAFLD. In this study, we used zebrafish for NAFLD research, and the results indicated the nutraceutical complex containing choline bitartrate, zinc, and vitamin E exhibits a pronounced hepatoprotective effect against NAFLD, resulting in reduced levels of alanine transaminase and

**Table 3. Primer Sequence Information.**

| gene | Primer Sequence Information | |
|---|---|---|
| *β-actin* | Forward | 5'-TCGAGCAGGAGATGGGAACC-3' |
| | Reverse | 5'-CTCGTGGATACCGCAAGATTC-3' |
| *acaca* | Forward | 5'-GGGCACAAAGACCGACAGAT-3' |
| | Reverse | 5'-GCTTCCTTAAAGCCTGGCGA-3' |
| *tbc1d1* | Forward | 5'-CCCACACCAGAGCAATCCTT-3' |
| | Reverse | 5'-CTTGAGAGGCCGTGTGGAAT-3' |
| *acsl1a* | Forward | 5'-TAACACCACTGAGACGTTGC-3' |
| | Reverse | 5'-ATGTGGTGATGGCAGCGAAT-3' |
| *fbp2* | Forward | 5'-CTGCGGAAAAACGAGGCAAG-3' |
| | Reverse | 5'-TGATGGTTCTCCATCTGAGATTCTT-3' |

**Table 4. Effects of nutraceutical complex on non-alcoholic fatty liver-related gene expression (*n*=3).**

| Group | concentration (µg/mL) | Relative expression of *acaca*# | Relative expression of *tbc1d1*# | Relative expression of *acsl1a*# | Relative expression of *fbp2*# |
|---|---|---|---|---|---|
| Normal control group (n=3) | – | 2.21±0.140ᵃ | 0.511±0.062ᵃ | 1.92±0.063ᵃ | 2.09±0.163ᵃ |
| Model control group (n=3) | – | 1.00±0.055 | 1.00±0.015 | 1.00±0.041 | 1.00±0.1294 |
| Nutraceutical complex containing zinc, choline bitartrate, and vitamin E (n=3) | 125 | 1.02±0.052 | 0.938±0.053 | 1.12±0.056 | 1.23±0.070 |
| | 250 | 1.93±0.061ᵃ | 0.744±0.037ᵃ | 1.60±0.074ᵃ | 1.58±0.159ᵃ |
| | 500 | 1.96±0.109ᵃ | 0.662±0.035ᵃ | 1.88±0.055ᵃ | 1.66±0.083ᵃ |

#Data are presented as mean±SE.

ᵃCompared with the control group of the model, $p < 0.05$.

aspartate aminotransferase activity, as well as enhanced hepatic fat transfer when compared with choline bitartrate alone intervention group. The mechanism behind the protective effects against NAFLD of the nutraceutical complex may upregulate the relative expression of TBC1D1 protein and downregulate the relative expression of ACACA, ACSL1 and FBP2 proteins. These findings suggest that the nutraceutical complex containing choline bitartrate, zinc, and vitamin E can improve NAFLD.

In the present study, we observed a more significant ameliorative effect produced by higher doses of choline bitartrate in a zebrafish model of NAFLD, suggesting a correlation between the dose of choline and its efficacy, and the exact efficacy relationship remains to be further investigated. In addition, to the best of our knowledge, no studies have explored the ameliorative effects of the nutraceutical complex containing choline, vitamin E, and zinc in NAFLD. The new nutraceutical complex, enriched with choline bitartrate, zinc, and vitamin E, produced a more favorable effect than the choline bitartrate intervention alone. For the possible mechanism of the combined action of the new nutraceutical complex, the results of the study indicated that the nutraceutical complex with a high content of choline and two other synergistic nutrients may downregulate the expression of ACACA protein. Acetyl coenzyme A carboxylase (ACACA) is a key rate-limiting enzyme that regulates the synthesis of fatty acids from scratch and plays an important role in lipid anabolism [40]. ACACA is expressed mainly in adipogenic tissues such as liver and adipose tissues, and the enzyme is in the cytoplasm of the cells [41]. ACACA promotes adipogenesis by stimulating fatty acid synthesis. ACC catalyzes the synthesis of malonyl coenzyme A from acetyl coenzyme A and carbonate, which is an important substrate for ab initio lipogenesis and inhibits fatty acid oxidation. Reducing hepatic steatosis by targeting the regulation of ACC and thereby reducing hepatic steatosis appears to be a potential therapy for treating NAFLD [42]. Zinc supplementation can significantly increase the activity of

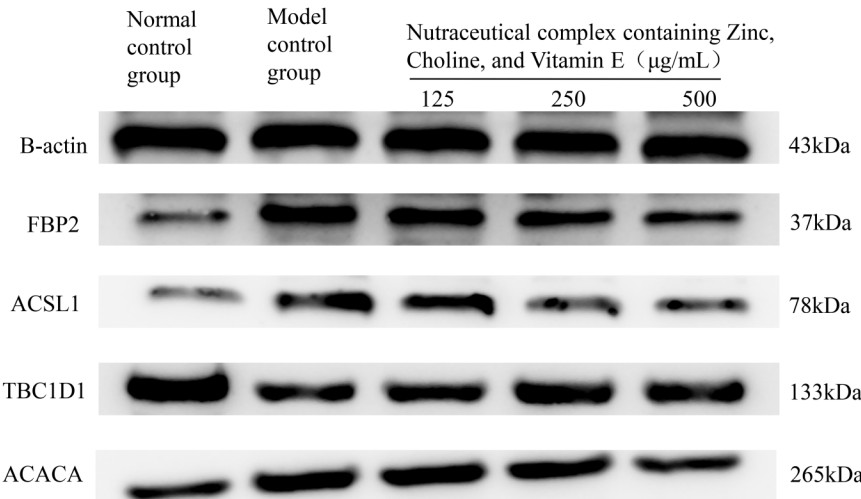

**Fig 4. The protein expression of *fbp2*, *acsl1*, *tbc1d1*, and *acaca*.**

acetyl coenzyme A carboxylase thereby increasing intramuscular fat deposition [43]. A recent study showed that aerobic exercise and vitamin E can significantly increase the expression of ACC enzymes and thus improve high-fat diet-induced NAFLD in rats [44]. Therefore, the nutraceutical complex may improve fatty liver by regulating ACC activity which may be attributed to adding vitamins and minerals.

The results of Polymerase Chain Reaction demonstrated that the nutraceutical complex may upregulate the relative expression of TBC1D1 protein and downregulate the relative expression of ACSL1 and FBP2 proteins. Long-chain lipoyl CoA synthetase (ACSL) is one of the key esterases involved in the ab initio synthesis of lipids and plays a key role in lipid biosynthesis and fatty acid degradation. ACSL1 was first discovered in ACSL subfamilies, which is associated with triglyceride synthesis and fatty acid uptake, which is widely present in the liver and adipocytes [45]. Overexpression of ACSL1 significantly increased triglyceride levels in tissues [46,47]. ACSL1a is a full-length ASCL1 cDNAs. In the current experiment, the nutraceutical complex containing choline bitartrate, zinc, and vitamin E presented protective efficacy against non-alcoholic fatty liver in zebrafish injected with acsl1a gene overexpression plasmid acsl1a-mRNA which suggested the potential to modulate triglyceride levels via regulating ACSL1a.

TBC1D1 is a Rab-GTPase-activating protein that is a downstream target of AMPK, and phosphorylation of TBC1D1 is required for glucose uptake by AMPK in skeletal muscle. It was shown that TBC1D1S231A-knock-in (KI) mice inhibit the phosphorylation of AMPK-TBC1D1 signaling through nutritional supplementation, block the AMPK-TBC1D1 pathway, increase the level of GTP-bound Rab2A, which in turn improves the protein stability of PPARγ and ultimately promotes the accumulation of intra- and extrahepatic lipids and leads to NAFLD in aged mice [48]. TBC1D1 can regulate lipid metabolism in different tissues through different mechanisms. Fructose-bisphosphate esterase (FBP2) acts as an activator of glucokinase and plays a role in maintaining hepatic metabolite homeostasis. Glucokinase expression and activity have been associated with type 2 diabetes mellitus and nonalcoholic fatty liver disease [49]. Fructose-2-phosphate kinase 6-phosphate/2-kinase/fructose-bisphosphate lyase (PFK2/FBP2) enhances the expression of glucokinase proteins when glucokinase and PFK2/FBP2 are co-expressed in hepatocytes [50]. The nutraceutical complex containing choline bitartrate, zinc, and vitamin E may play a role in regulating the proteins involved, but further research is needed.

Although there are several emerging therapeutic approaches for the treatment of NAFLD, several formulations were also reported to have the potential to treat NAFLD [51,52]. In our study, we explored the potential effect, and the underlying mechanisms of the nutraceutical complex containing choline bitartrate, zinc, and vitamin E on the amelioration of

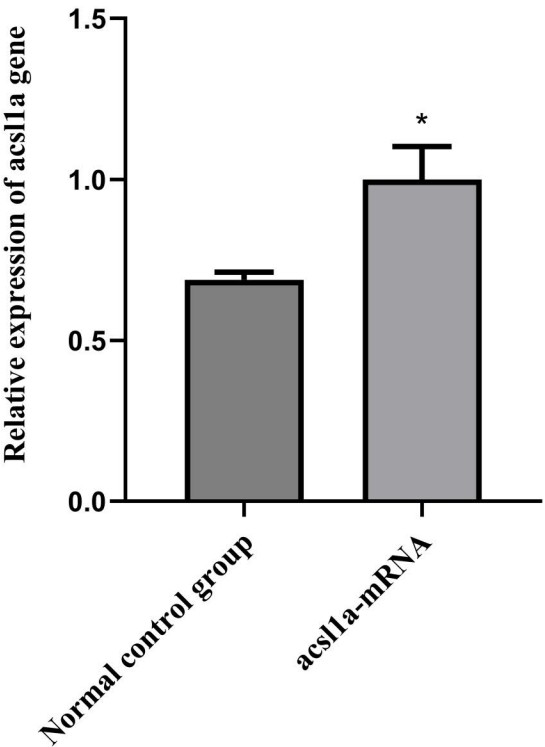

**Fig 5. Relative expression of *acsl1a* gene.** *Comparison with normal control group, $p < 0.05$.

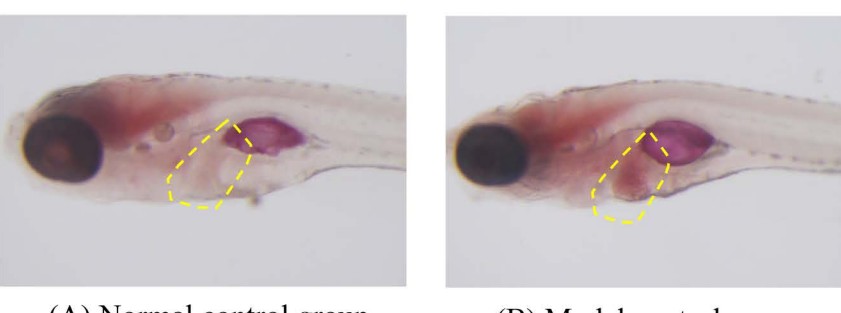

(A) Normal control group          (B) Model control group

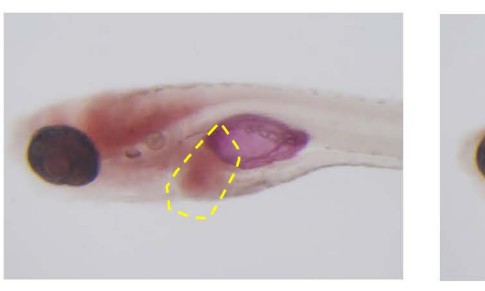

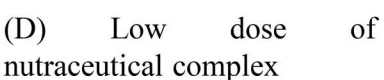

(D) Low dose of nutraceutical complex

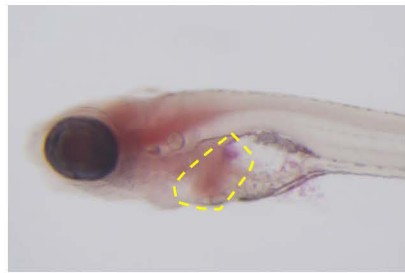

(E) Middle dose of nutraceutical complex

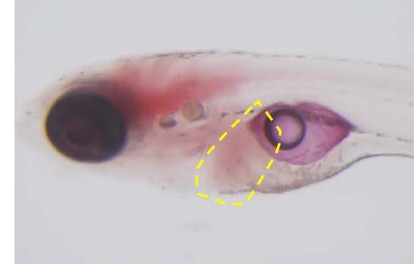

(F) High dose of nutraceutical complex

**Fig 6. Typical plot of zebrafish liver fat staining intensity after sample treatment.** The yellow dashed box is the liver.

NAFLD. A single component such as choline bitartrate indeed has effects on the improvement of NAFLD, but the effect of the nutraceutical complex containing choline bitartrate, zinc, and vitamin E surpasses that of choline bitartrate alone in the current trial. The results provided insight into the strategy based on dietary supplementation to improve NAFLD. The present study still has some limitations. Firstly, considering the differences between zebrafish and humans, the application of the results of the zebrafish experiment to population-based clinical studies remains to be confirmed by many studies. Secondly, although transcriptomics has identified many differential genes, other unattended genes may still have a role in NAFLD, which needs to be further explored by future studies. Finally, the present study is based on zebrafish, which has a limited sample size and needs to be confirmed by further large samples of relevant animal studies in the future.

## Conclusions

In summary, the results from this study indicated that choline bitartrate combined with zinc citrate, and dl-α-Tocopheryl acetate can improve NAFLD. The mechanism behind the effect of choline bitartrate combined with zinc citrate, and dl-α-Tocopheryl acetate may be an upregulation of the relative expression of TBC1D1 protein and the downregulation of the relative expression of ACACA, ACSL1, and FBP2 proteins. The results of this study provide evidence for applying the nutraceutical complex in the improvement of NAFLD and provide references to the development of dietary supplements for NAFLD.

## Supporting information

**S1 File.**   **S1 Fig.** Wayne diagrams of differential genes. CB: Choline bitartrate control group; FF: Nutraceutical complex containing Zinc, Choline bitartrate, and Vitamin E. **S2 Fig.** GO annotation analysis of the choline bitartrate differential gene. (A) GO classification of the choline bitartrate differential gene. (B) GO Biological process enrichment bubble diagram. (C) GO Molecular function enrichment bubble diagram. (D) GO Cellular component enrichment bubble map. **S3 Fig.** GO annotation analysis of functional formulation differential genes. (A) GO classification of the Functional formulation differential gene. (B) GO Biological process enrichment bubble diagram. (C) GO Cellular component enrichment bubble map. (D) GO Molecular function enrichment bubble diagram. **S4 Fig.** Differential gene KEGG enrichment. (A) KEGG enrichment of the choline bitartrate differential gene. (B) KEGG enrichment of the functional formulation differential gene. **S1 Table.** The concentration of nutraceutical complex for the protective effects against non-alcoholic fatty liver. **S2 Table.** The protective effect of nutraceutical complex on NAFLD (alanine transaminase activity). **S3 Table.** The protective effect of nutraceutical complex on NAFLD (Aspartate aminotransferase activity). **S4 Table.** The pathways and differential genes involved in the nutraceutical complex intervention group. **S5 Table.** Effect of nutraceutical complex on the expression of NAFLD-associated proteins (n = 3). **S6 Table.** Results of the acsl1a-mRNA dose mapping experiment (n = 30). **S7 Table.** Concentration of total RNA and A260/A280 ratio (n = 3). **S8 Table.** Primer Sequence Information. **S9 Table.** Results of experiments to evaluate the protective efficacy of samples against NAFLD (n = 10).
(DOCX)

## Acknowledgments

The authors thank Zhen-Hua Wu and Yin-Li Wang for fish husbandry.

## Author contributions

**Conceptualization:** Bingbing Cao, Jiali zhou.

**Data curation:** Bingbing Cao, Rui Wang.

**Formal analysis:** Bingbing Cao.

**Funding acquisition:** Chunqi Li.

**Investigation:** Jiali zhou, Xiaoqing Li, Rui Wang.

**Methodology:** Bo Xia.

**Project administration:** Yiqiao Xu, Chunqi Li.

**Resources:** Bo Xia, Yiqiao Xu.

**Software:** Bo Xia, Yiqiao Xu.

**Supervision:** Yiqiao Xu.

**Validation:** Xiaoqing Li, Rui Wang, Yiqiao Xu.

**Visualization:** Rui Wang.

**Writing – original draft:** Bingbing Cao, Jiali zhou.

**Writing – review & editing:** Chunqi Li.

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
