## [Decision Letter · Decision Letter 0]

13 Nov 2024

PONE-D-24-40651Therapeutic Potential of a Choline-Zinc-Vitamin E Nutraceutical Complex in Ameliorating Thioacetamide-Induced Nonalcoholic Fatty Liver Pathology in ZebrafishPLOS ONE

Dear Dr. zhou,

Thank you for submitting your manuscript to PLOS ONE. After careful consideration, we feel that it has merit but does not fully meet PLOS ONE’s publication criteria as it currently stands. Therefore, we invite you to submit a revised version of the manuscript that addresses the points raised during the review process.

We look forward to receiving your revised manuscript.

Kind regards,

Fakhar Islam, PhD

Academic Editor

PLOS ONE

Journal Requirements:

3. Thank you for stating the following financial disclosure: [Sanofi funded the study]. Please state what role the funders took in the study. If the funders had no role, please state: "The funders had no role in study design, data collection and analysis, decision to publish, or preparation of the manuscript." If this statement is not correct you must amend it as needed. Please include this amended Role of Funder statement in your cover letter; we will change the online submission form on your behalf.

4. Thank you for stating the following in the Competing Interests section: [Xiaoqing Li and Rui Wang are currently employees of Opella. Bingbing Cao, Jiali Zhou, Xia Bo, Yiqiao Xu and Chunqi Li are employees of Hunter Biotechnology, Inc. which performed the study by order of Opella.]. Please confirm that this does not alter your adherence to all PLOS ONE policies on sharing data and materials, by including the following statement: "This does not alter our adherence to PLOS ONE policies on sharing data and materials.” (as detailed online in our guide for authors http://journals.plos.org/plosone/s/competing-interests). If there are restrictions on sharing of data and/or materials, please state these. Please note that we cannot proceed with consideration of your article until this information has been declared. Please include your updated Competing Interests statement in your cover letter; we will change the online submission form on your behalf.

5. PLOS ONE now requires that authors provide the original uncropped and unadjusted images underlying all blot or gel results reported in a submission’s figures or Supporting Information files. This policy and the journal’s other requirements for blot/gel reporting and figure preparation are described in detail at https://journals.plos.org/plosone/s/figures#loc-blot-and-gel-reporting-requirements and https://journals.plos.org/plosone/s/figures#loc-preparing-figures-from-image-files. When you submit your revised manuscript, please ensure that your figures adhere fully to these guidelines and provide the original underlying images for all blot or gel data reported in your submission. See the following link for instructions on providing the original image data: https://journals.plos.org/plosone/s/figures#loc-original-images-for-blots-and-gels. In your cover letter, please note whether your blot/gel image data are in Supporting Information or posted at a public data repository, provide the repository URL if relevant, and provide specific details as to which raw blot/gel images, if any, are not available. Email us at plosone@plos.org if you have any questions.

6. PLOS requires an ORCID iD for the corresponding author in Editorial Manager on papers submitted after December 6th, 2016. Please ensure that you have an ORCID iD and that it is validated in Editorial Manager. To do this, go to ‘Update my Information’ (in the upper left-hand corner of the main menu), and click on the Fetch/Validate link next to the ORCID field. This will take you to the ORCID site and allow you to create a new iD or authenticate a pre-existing iD in Editorial Manager.

7. Please include a separate caption for each figure in your manuscript.

8. Please include a copy of Tables 1-4 which you refer to in your text on your manuscript.

9. Please include captions for your Supporting Information files at the end of your manuscript, and update any in-text citations to match accordingly. Please see our Supporting Information guidelines for more information: http://journals.plos.org/plosone/s/supporting-information .

Additional Editor Comments:

Reviewer-1

I am writing to submit my review of the manuscript entitled " Therapeutic Potential of a Choline-Zinc-Vitamin E Nutraceutical Complex in Ameliorating Thioacetamide-Induced Nonalcoholic Fatty Liver Pathology in Zebrafish" for your consideration. Overall, I find the manuscript's findings intriguing and the information provided useful for researchers and academia. The article has the potential to make a significant contribution to the related discipline.

However, I have some concerns regarding the clarity, detail, and precision of different sections, which I outline below:

I recommend that the authors address these concerns and provide a revised version of the manuscript for further consideration

· L92- Reduce the redundancy and make it more precise. Nonalcoholic fatty liver disease (NAFLD) has been troubling almost one-quarter of global population for several years and its prevalence continues to rise.

· L 98- Please rephrase it. To address NAFLD, operational management approaches encompass lifestyle modifications, weight reduction, and the potential utilization of various biomolecules, including caffeine and polyphenols, which have shown promise in ameliorating the condition.

· L 102- Make it more clear. However, there is no accurate evidence on their treatment's optimal dose and duration.

· L 105- These lines need to be improved. Medical interventions like T2DM drugs (Glucagon-like peptide-1 receptor agonists and Liraglutide), lipid-lowering drugs (Statins and Ezetimibe) and antihypertensive drugs are reported to have a certain effect in ameliorating NAFLD while they tend to cause a series of side effects such as diarrhea, vomiting, constipation and loss of appetite.

· L 109 This line should be precised and easily readable-The neurotransmitter acetylcholine is composed of choline, which is a constituent of mitochondrial membranes and cell membranes.

· L 122 The concet of these lines is not clear- As a lipid-soluble antioxidant, vitamin E prevents the spread of free radicals.

· L 136-138 These lines should be more clear - A new nutraceutical complex of choline bitartrate, zinc citrate, and dl-α-Tocopheryl acetate is developed to help improve the NAFLD.

· L 172 This statement is not clear it should by specify - Oil Red O (lot number: SHBN4926, Sigma, USA).

· L 211-214 Break it into two sentences for more clarification. Interfering with intracellular nucleus RNA transfer, affecting protein synthesis and enzyme activity, increasing DNA synthesis and mitosis in the nucleus of hepatocytes, promoting the development of cirrhosis.

· L 370-371 Reconsider it. The protective effects of the nutraceutical complex at concentrations of 250 and 500 μg/mL are superior to those of choline bitartrate at concentrations of 162 and 316 μg/mL when the content of choline bitartrate is equivalent (p < 0.001).

· L 386-387 Break the sentence for better comparison -the liver cells in the model control group showed loose struc ture with numerous lipid vacuoles; the positive control group at a concentration of 50.0 μg/mL showed compact liver cell structure similar to the normal control group, with a significant reduction in lipid vacuoles compared to the model control group, indicating a significant hepatoprotective effect of the positive control group.

· L 449-352 Avoid too long sentences - The concentration and purity of total RNA were determined using a UV-Vis spectrophotometer. 2.00 μg of total RNA from zebrafish samples was taken and 20.0 μL of cDNA was synthesized according to the instructions of the cDNA First Strand Synthesis Kit, and the expression of β-actin and acsl1a genes was detected by q-PCR.

· L 427- 428 Minor grammatically mistake - The KEGG Pathway enrichment bubble diagram is presented in Figure S4.

· L 443-445 You haven’t mentioned the units if it is measured by spectrophotometer- The ratio of A260/A280 was between 1.8-2.2, 444 indicating that the extracted the quality of fish total RNA is good and can be used in 445 subsequent q-PCR experiments.

· The introduction & discussion section could be improved by providing more context and background from following references,

· doi: https://doi.org/10.1016/j.phymed.2024.155353

· doi: https://doi.org/10.1016/j.phymed.2022.154090

· doi: https://doi.org/10.1016/j.pacs.2023.100569

· doi: 10.1620/tjem.2022.J083

· https://doi.org/10.7150/thno.42539

Italic all the scientific names,

· Remove grammatical mistakes

· Need to rewrite the conclusion

§ Recheck Legends description is as per figure number and discussion-

§ I urge the authors to improve the English language for better flow of literature.

§ Please check reference style throughout MS

Reviewer-2

• Mention full forms of all abbreviations at its first mention.

• Why Choline-Zinc-Vitamin E together? Justify.

• Mention the limitation and future prospects in conclusion.

• Did not found any table at your submission.

• Figure no and Result figure has not matched properly.

Reviewers' comments:

Reviewer's Responses to Questions

**Comments to the Author**

1. Is the manuscript technically sound, and do the data support the conclusions?

Reviewer #1: Yes

Reviewer #2: Yes

2. Has the statistical analysis been performed appropriately and rigorously? 

Reviewer #1: Yes

Reviewer #2: I Don't Know

3. Have the authors made all data underlying the findings in their manuscript fully available?

Reviewer #1: Yes

Reviewer #2: No

4. Is the manuscript presented in an intelligible fashion and written in standard English?

Reviewer #1: Yes

Reviewer #2: Yes

5. Review Comments to the Author

Reviewer #1: I am writing to submit my review of the manuscript entitled " Therapeutic Potential of a Choline-Zinc-Vitamin E Nutraceutical Complex in Ameliorating Thioacetamide-Induced Nonalcoholic Fatty Liver Pathology in Zebrafish" for your consideration. Overall, I find the manuscript's findings intriguing and the information provided useful for researchers and academia. The article has the potential to make a significant contribution to the related discipline.

However, I have some concerns regarding the clarity, detail, and precision of different sections, which I outline below:

I recommend that the authors address these concerns and provide a revised version of the manuscript for further consideration

· L92- Reduce the redundancy and make it more precise. Nonalcoholic fatty liver disease (NAFLD) has been troubling almost one-quarter of global population for several years and its prevalence continues to rise.

· L 98- Please rephrase it. To address NAFLD, operational management approaches encompass lifestyle modifications, weight reduction, and the potential utilization of various biomolecules, including caffeine and polyphenols, which have shown promise in ameliorating the condition.

· L 102- Make it more clear. However, there is no accurate evidence on their treatment's optimal dose and duration.

· L 105- These lines need to be improved. Medical interventions like T2DM drugs (Glucagon-like peptide-1 receptor agonists and Liraglutide), lipid-lowering drugs (Statins and Ezetimibe) and antihypertensive drugs are reported to have a certain effect in ameliorating NAFLD while they tend to cause a series of side effects such as diarrhea, vomiting, constipation and loss of appetite.

· L 109 This line should be precised and easily readable-The neurotransmitter acetylcholine is composed of choline, which is a constituent of mitochondrial membranes and cell membranes.

· L 122 The concet of these lines is not clear- As a lipid-soluble antioxidant, vitamin E prevents the spread of free radicals.

· L 136-138 These lines should be more clear - A new nutraceutical complex of choline bitartrate, zinc citrate, and dl-α-Tocopheryl acetate is developed to help improve the NAFLD.

· L 172 This statement is not clear it should by specify - Oil Red O (lot number: SHBN4926, Sigma, USA).

· L 211-214 Break it into two sentences for more clarification. Interfering with intracellular nucleus RNA transfer, affecting protein synthesis and enzyme activity, increasing DNA synthesis and mitosis in the nucleus of hepatocytes, promoting the development of cirrhosis.

· L 370-371 Reconsider it. The protective effects of the nutraceutical complex at concentrations of 250 and 500 μg/mL are superior to those of choline bitartrate at concentrations of 162 and 316 μg/mL when the content of choline bitartrate is equivalent (p < 0.001).

· L 386-387 Break the sentence for better comparison -the liver cells in the model control group showed loose struc ture with numerous lipid vacuoles; the positive control group at a concentration of 50.0 μg/mL showed compact liver cell structure similar to the normal control group, with a significant reduction in lipid vacuoles compared to the model control group, indicating a significant hepatoprotective effect of the positive control group.

· L 449-352 Avoid too long sentences - The concentration and purity of total RNA were determined using a UV-Vis spectrophotometer. 2.00 μg of total RNA from zebrafish samples was taken and 20.0 μL of cDNA was synthesized according to the instructions of the cDNA First Strand Synthesis Kit, and the expression of β-actin and acsl1a genes was detected by q-PCR.

· L 427- 428 Minor grammatically mistake - The KEGG Pathway enrichment bubble diagram is presented in Figure S4.

· L 443-445 You haven’t mentioned the units if it is measured by spectrophotometer- The ratio of A260/A280 was between 1.8-2.2, 444 indicating that the extracted the quality of fish total RNA is good and can be used in 445 subsequent q-PCR experiments.

· The introduction & discussion section could be improved by providing more context and background from following references,

· doi: https://doi.org/10.1016/j.phymed.2024.155353

· doi: https://doi.org/10.1016/j.phymed.2022.154090

· doi: https://doi.org/10.1016/j.pacs.2023.100569

· doi: 10.1620/tjem.2022.J083

· https://doi.org/10.7150/thno.42539

Italic all the scientific names,

· Remove grammatical mistakes

· Need to rewrite the conclusion

§ Recheck Legends description is as per figure number and discussion-

§ I urge the authors to improve the English language for better flow of literature.

§ Please check reference style throughout MS

Reviewer #2: • Mention full forms of all abbreviations at its first mention.

• Why Choline-Zinc-Vitamin E together? Justify.

• Mention the limitation and future prospects in conclusion.

• Did not found any table at your submission.

• Figure no and Result figure has not matched properly.

6. PLOS authors have the option to publish the peer review history of their article (what does this mean? ). If published, this will include your full peer review and any attached files.

**Do you want your identity to be public for this peer review?** For information about this choice, including consent withdrawal, please see our Privacy Policy .

Reviewer #1: **Yes: ** Dr. Muhammad Afzaal

Reviewer #2: No

---

## [Author Response · Author response to Decision Letter 1]

18 Dec 2024

Response to “Journal Requirements”:

Response: Thank you very much for your suggestions, we have revised the manuscript with formatting adjustments based on the journal template.

Response: Thank you very much for your suggestions, we have added a description of the humanistic and ethical aspects of animal experimentation in the methodology section.

3. Thank you for stating the following financial disclosure: [Sanofi funded the study]. Please state what role the funders took in the study. If the funders had no role, please state: "The funders had no role in study design, data collection and analysis, decision to publish, or preparation of the manuscript." If this statement is not correct you must amend it as needed. Please include this amended Role of Funder statement in your cover letter; we will change the online submission form on your behalf.

Response: Thank you very much for your suggestions, we have revised the manuscript.

4. Thank you for stating the following in the Competing Interests section: [Xiaoqing Li and Rui Wang are currently employees of Opella. Bingbing Cao, Jiali Zhou, Xia Bo, Yiqiao Xu and Chunqi Li are employees of Hunter Biotechnology, Inc. which performed the study by order of Opella.]. Please confirm that this does not alter your adherence to all PLOS ONE policies on sharing data and materials, by including the following statement: "This does not alter our adherence to PLOS ONE policies on sharing data and materials.” (as detailed online in our guide for authors http://journals.plos.org/plosone/s/competing-interests). If there are restrictions on sharing of data and/or materials, please state these. Please note that we cannot proceed with consideration of your article until this information has been declared. Please include your updated Competing Interests statement in your cover letter; we will change the online submission form on your behalf.

Response: Thank you very much for your suggestions, we have added this in the cover letter.

5. PLOS ONE now requires that authors provide the original uncropped and unadjusted images underlying all blot or gel results reported in a submission’s figures or Supporting Information files. This policy and the journal’s other requirements for blot/gel reporting and figure preparation are described in detail at https://journals.plos.org/plosone/s/figures#loc-blot-and-gel-reporting-requirements and https://journals.plos.org/plosone/s/figures#loc-preparing-figures-from-image-files. When you submit your revised manuscript, please ensure that your figures adhere fully to these guidelines and provide the original underlying images for all blot or gel data reported in your submission. See the following link for instructions on providing the original image data: https://journals.plos.org/plosone/s/figures#loc-original-images-for-blots-and-gels. In your cover letter, please note whether your blot/gel image data are in Supporting Information or posted at a public data repository, provide the repository URL if relevant, and provide specific details as to which raw blot/gel images, if any, are not available. Email us at plosone@plos.org if you have any questions.

Response: Thank you very much for your suggestions, we have added the original uncropped and unadjusted images underlying all blot in the end of Supporting Information.

6. PLOS requires an ORCID iD for the corresponding author in Editorial Manager on papers submitted after December 6th, 2016. Please ensure that you have an ORCID iD and that it is validated in Editorial Manager. To do this, go to ‘Update my Information’ (in the upper left-hand corner of the main menu), and click on the Fetch/Validate link next to the ORCID field. This will take you to the ORCID site and allow you to create a new iD or authenticate a pre-existing iD in Editorial Manager.

Response: Thank you very much for your suggestions, we have registered (https://orcid.org/0009-0009-8025-7896).

7. Please include a separate caption for each figure in your manuscript.

Response: Thank you very much for your suggestions, we have added them.

8. Please include a copy of Tables 1-4 which you refer to in your text on your manuscript.

Response: Thank you very much for your suggestions, we have added them.

Response: Thank you very much for your suggestions, we have added them.

Response to Reviewer-1:

1、I am writing to submit my review of the manuscript entitled " Therapeutic Potential of a Choline-Zinc-Vitamin E Nutraceutical Complex in Ameliorating Thioacetamide-Induced Nonalcoholic Fatty Liver Pathology in Zebrafish" for your consideration. Overall, I find the manuscript's findings intriguing and the information provided useful for researchers and academia. The article has the potential to make a significant contribution to the related discipline.

However, I have some concerns regarding the clarity, detail, and precision of different sections, which I outline below:

I recommend that the authors address these concerns and provide a revised version of the manuscript for further consideration.

Response: Thank you very much for your work and valuable suggestions. We have revised the manuscript one by one according to your comments, and hope that the revised manuscript can satisfy you, thank you again for your comments!

2、L92- Reduce the redundancy and make it more precise. Nonalcoholic fatty liver disease (NAFLD) has been troubling almost one-quarter of global population for several years and its prevalence continues to rise.

Response: Thank you very much for your comments, we have reviewed the relevant literature and re-described the content to give specific values.

3、L 98- Please rephrase it. To address NAFLD, operational management approaches encompass lifestyle modifications, weight reduction, and the potential utilization of various biomolecules, including caffeine and polyphenols, which have shown promise in ameliorating the condition.

Response: Thank you very much for your comments, we have rewritten this.

4、L 102- Make it more clear. However, there is no accurate evidence on their treatment's optimal dose and duration.

Response: Thank you very much for your comments, we have deleted this.

5、L 105- These lines need to be improved. Medical interventions like T2DM drugs (Glucagon-like peptide-1 receptor agonists and Liraglutide), lipid-lowering drugs (Statins and Ezetimibe) and antihypertensive drugs are reported to have a certain effect in ameliorating NAFLD while they tend to cause a series of side effects such as diarrhea, vomiting, constipation and loss of appetite.

Response: Thank you very much for your comments, we have rewritten this.

6、L 109 This line should be precised and easily readable-The neurotransmitter acetylcholine is composed of choline, which is a constituent of mitochondrial membranes and cell membranes.

Response: Thank you very much for your comments, we have rewritten this.

7、L 122 The concet of these lines is not clear- As a lipid-soluble antioxidant, vitamin E prevents the spread of free radicals.

Response: Thank you very much for your comments, we have rewritten this.

8、L 136-138 These lines should be more clear - A new nutraceutical complex of choline bitartrate, zinc citrate, and dl-α-Tocopheryl acetate is developed to help improve the NAFLD.

Response: Thank you very much for your comments, we have deleted this.

9、L 172 This statement is not clear it should by specify - Oil Red O (lot number: SHBN4926, Sigma, USA).

Response: Thank you very much for your comments, we have revised this.

10、L 211-214 Break it into two sentences for more clarification. Interfering with intracellular nucleus RNA transfer, affecting protein synthesis and enzyme activity, increasing DNA synthesis and mitosis in the nucleus of hepatocytes, promoting the development of cirrhosis.

Response: Thank you very much for your comments, we have revised this.

11、L 370-371 Reconsider it. The protective effects of the nutraceutical complex at concentrations of 250 and 500 μg/mL are superior to those of choline bitartrate at concentrations of 162 and 316 μg/mL when the content of choline bitartrate is equivalent (p < 0.001).

Response: Thank you very much for your comments, we rewrite the sentences to avoid ambiguity.

12、L 386-387 Break the sentence for better comparison -the liver cells in the model control group showed loose structure with numerous lipid vacuoles; the positive control group at a concentration of 50.0 μg/mL showed compact liver cell structure similar to the normal control group, with a significant reduction in lipid vacuoles compared to the model control group, indicating a significant hepatoprotective effect of the positive control group.

Response: Thank you very much for your comments, we have revised this.

13、L 449-352 Avoid too long sentences - The concentration and purity of total RNA were determined using a UV-Vis spectrophotometer. 2.00 μg of total RNA from zebrafish samples was taken and 20.0 μL of cDNA was synthesized according to the instructions of the cDNA First Strand Synthesis Kit, and the expression of β-actin and acsl1a genes was detected by q-PCR.

Response: Thank you very much for your comments, we have revised this and deleted the redundant part of the sentence.

14、L 427- 428 Minor grammatically mistake - The KEGG Pathway enrichment bubble diagram is presented in Figure S4.

Response: Thank you very much for your comments, we have revised this.

15、L 443-445 You haven’t mentioned the units if it is measured by spectrophotometer- The ratio of A260/A280 was between 1.8-2.2, 444 indicating that the extracted the quality of fish total RNA is good and can be used in 445 subsequent q-PCR experiments.

Response: Thank you very much for your comments, The unit of the RNA spectrophotometer is usually the absorbance, denoted by “A”, which is in units of 1 (unitless). There are usually no units.

16、The introduction & discussion section could be improved by providing more context and background from following references,

· doi: https://doi.org/10.1016/j.phymed.2024.155353 IF: 6.7 Q1

· doi: https://doi.org/10.1016/j.phymed.2022.154090 IF: 6.7 Q1

· doi: https://doi.org/10.1016/j.pacs.2023.100569 IF: 7.1 Q1

· doi: 10.1620/tjem.2022.J083 IF: 1.7 Q2

· https://doi.org/10.7150/thno.42539 IF: 12.4 Q1

Italic all the scientific names,

Response: Thank you very much for your comments; we have read this literature and selected appropriate content to cite in the introduction and discussion.

17、Remove grammatical mistakes

Response: Thank you very much for your comments, we have checked this.

18、Need to rewrite the conclusion

Response: Thank you very much for your comments, we have revised this.

19、§ Recheck Legends description is as per figure number and discussion-

Response: Thank you very much for your comments, we have checked this.

20、§ I urge the authors to improve the English language for better flow of literature.

Response: Thank you very much for your comments, we have given the manuscript to native English-speaking international students for a careful grammar check.

21、§ Please check reference style throughout MS

Response: Thank you very much for your comments, we have checked this.

Response to Reviewer-2:

1、• Mention full forms of all abbreviations at its first mention.

Response: Thank you very much for your comments, we have checked this in the manuscript.

2、• Why Choline-Zinc-Vitamin E together? Justify.

Response: Thank you very much for your comments. Dietary nutrient supplements and natural herbal ingredients may be critical in improving NAFLD; considering that choline, vitamin E, and zinc all have some hepatoprotective potential. Therefore, the present study was designed to explore the potential effects of choline bitartrate, combined with zinc citrate, and dl-α-Tocopheryl acetate on improving NAFLD based on zebrafish. The results of this study may provide evidence for applying the nutraceutical complex in the improvement of NAFLD and provide references to the development of dietary supplements for NAFLD.

3、• Mention the limitation and future prospects in conclusion.

Response: Thank you very much for your comments; we have added relevant content to the manuscript.

4、• Did not found any table at your submission.

Response: we have added tables to the manuscript.

5、• Figure no and Result figure has not matched properly.

Response: Thank you very much for your comments, we have corrected and modified this error.

---

## [Decision Letter · Decision Letter 1]

22 Jan 2025

PONE-D-24-40651R1Therapeutic potential of a choline-zinc-vitamin e nutraceutical complex in ameliorating thioacetamide-induced nonalcoholic fatty liver pathology in zebrafishPLOS ONE

Dear Dr. zhou,

Thank you for submitting your manuscript to PLOS ONE. After careful consideration, we feel that it has merit but does not fully meet PLOS ONE’s publication criteria as it currently stands. Therefore, we invite you to submit a revised version of the manuscript that addresses the points raised during the review process.

We look forward to receiving your revised manuscript.

Kind regards,

Fakhar Islam, PhD

Academic Editor

PLOS ONE

Journal Requirements:

Additional Editor Comments:

• In “Hematoxylin and Eosin Staining, and Oil Red O Staining” paragraph, kindly mention the full form of H&E.

• In line 213, kindly mention the sample name.

• In result S1, S2, S3, S4 table and Figure means which table and figure? In supporting information no figure and table found. Kindly check.

• In Table 1 , which substrate’s concentration? Please mention.

Reviewers' comments:

Reviewer's Responses to Questions

**Comments to the Author**

1. If the authors have adequately addressed your comments raised in a previous round of review and you feel that this manuscript is now acceptable for publication, you may indicate that here to bypass the “Comments to the Author” section, enter your conflict of interest statement in the “Confidential to Editor” section, and submit your "Accept" recommendation.

Reviewer #1: All comments have been addressed

Reviewer #2: All comments have been addressed

2. Is the manuscript technically sound, and do the data support the conclusions?

Reviewer #1: Yes

Reviewer #2: Yes

3. Has the statistical analysis been performed appropriately and rigorously? 

Reviewer #1: Yes

Reviewer #2: Yes

4. Have the authors made all data underlying the findings in their manuscript fully available?

Reviewer #1: Yes

Reviewer #2: No

5. Is the manuscript presented in an intelligible fashion and written in standard English?

Reviewer #1: Yes

Reviewer #2: Yes

6. Review Comments to the Author

Reviewer #1: Dear Editor, The authors have responded to all the raised queries, therefore, the Manuscript can be accepted

Reviewer #2: • In “Hematoxylin and Eosin Staining, and Oil Red O Staining” paragraph, kindly mention the full form of H&E.

• In line 213, kindly mention the sample name.

• In result S1, S2, S3, S4 table and Figure means which table and figure? In supporting information no figure and table found. Kindly check.

• In Table 1 , which substrate’s concentration? Please mention.

7. PLOS authors have the option to publish the peer review history of their article (what does this mean? ). If published, this will include your full peer review and any attached files.

**Do you want your identity to be public for this peer review?** For information about this choice, including consent withdrawal, please see our Privacy Policy .

Reviewer #1: **Yes: ** Dr. Muhammad Afzaal

Reviewer #2: No

---

## [Author Response · Author response to Decision Letter 2]

4 Feb 2025

Response to “Journal Requirements”:

Response: Thank you very much for your reminder. We have checked the references of the manuscript, and all references are correct and complete, and there are no withdrawn references.

Response to Reviewer-2:

1、In “Hematoxylin and Eosin Staining, and Oil Red O Staining” paragraph, kindly mention the full form of H&E.

Response: Thank you very much for your careful and valuable comments, we have added the full form of H&E in the appropriate section.

2、In line 213, kindly mention the sample name.

Response: Thank you very much for your careful comments, we have made changes accordingly. We change “ the MTC of the sample ” to “ the Maximum tolerated concentration (MTC) of the nutraceutical complex containing choline bitartrate, zinc citrate, and dl-α-Tocopheryl acetate ”.

3、In result S1, S2, S3, S4 table and Figure means which table and figure? In supporting information no figure and table found. Kindly check.

Response: Thank you for your careful comments, we have re-checked and revised them carefully to ensure that the citations in the body of the text correspond to the content of the supplementary material. In the support information section, we have also added images and tables.

4、In Table 1 , which substrate’s concentration? Please mention.

Response: Thank you very much for your comments, we have added superscripts and added instructions below the table as follows:

In the positive control group, the concentration represents the concentration of polyene phosphatidylcholine. In addition, the nutraceutical complex group containing zinc, choline bitartrate, and vitamin E and the choline bitartrate group each had three concentration levels.

---

## [Decision Letter · Decision Letter 2]

22 Apr 2025

Therapeutic potential of a choline-zinc-vitamin e nutraceutical complex in ameliorating thioacetamide-induced nonalcoholic fatty liver pathology in zebrafish

PONE-D-24-40651R2

Dear Dr. zhou,

We’re pleased to inform you that your manuscript has been judged scientifically suitable for publication and will be formally accepted for publication once it meets all outstanding technical requirements.

Kind regards,

Fahrul Nurkolis

Academic Editor

PLOS ONE

Additional Editor Comments (optional):

Accept in the present form!

Reviewers' comments:

Reviewer's Responses to Questions

**Comments to the Author**

1. If the authors have adequately addressed your comments raised in a previous round of review and you feel that this manuscript is now acceptable for publication, you may indicate that here to bypass the “Comments to the Author” section, enter your conflict of interest statement in the “Confidential to Editor” section, and submit your "Accept" recommendation.

Reviewer #2: All comments have been addressed

2. Is the manuscript technically sound, and do the data support the conclusions?

Reviewer #2: Yes

3. Has the statistical analysis been performed appropriately and rigorously? 

Reviewer #2: Yes

4. Have the authors made all data underlying the findings in their manuscript fully available?

Reviewer #2: Yes

5. Is the manuscript presented in an intelligible fashion and written in standard English?

Reviewer #2: Yes

6. Review Comments to the Author

Reviewer #2: (No Response)

7. PLOS authors have the option to publish the peer review history of their article (what does this mean? ). If published, this will include your full peer review and any attached files.

**Do you want your identity to be public for this peer review?** For information about this choice, including consent withdrawal, please see our Privacy Policy .

Reviewer #2: **Yes: ** Sunanda Biswas

---

## [Editor Report · Acceptance letter]

PONE-D-24-40651R2

PLOS ONE

Dear Dr. zhou,

I'm pleased to inform you that your manuscript has been deemed suitable for publication in PLOS ONE. Congratulations! Your manuscript is now being handed over to our production team.

Kind regards,

on behalf of

Dr. Fahrul Nurkolis

Academic Editor

PLOS ONE